# Isolation, Physiological Characterization, and Antibiotic Susceptibility Testing of Fast-Growing Bacteria from the Sea-Affected Temporary Meltwater Ponds in the Thala Hills Oasis (Enderby Land, East Antarctica)

**DOI:** 10.3390/biology11081143

**Published:** 2022-07-29

**Authors:** Volha Akulava, Uladzislau Miamin, Katsiaryna Akhremchuk, Leonid Valentovich, Andrey Dolgikh, Volha Shapaval

**Affiliations:** 1Faculty of Science and Technology, Norwegian University of Life Sciences, 1432 Ås, Norway; volha.shapaval@nmbu.no; 2Faculty of Biology, Belarusian State University, 220030 Minsk, Belarus; vladmiamin@mail.ru (U.M.); valentovich@bio.bsu.by (L.V.); 3Scientific and Practical Center of the National Academy of Sciences of Belarus for Bioresources, 220072 Minsk, Belarus; 4Institute of Microbiology, National Academy of Sciences of Belarus, 220141 Minsk, Belarus; katerina_akhr@bio.bsu.by; 5Institute of Geography, Russian Academy of Sciences, 119017 Moscow, Russia; dolgikh@igras.ru

**Keywords:** meltwater ponds, Antarctic bacteria, 16S rRNA gene sequencing, plate-based assays, thermotolerance, enzymatic activity, antibiotic resistance

## Abstract

**Simple Summary:**

The characterization of microbial communities from Antarctic temporary meltwater ponds is limited, while they could serve as a source of biotechnologically interesting microorganisms. In this study, we characterized a set of bacteria isolated from the sea-affected temporary meltwater ponds in the East Antarctica area of the Vecherny region of the Thala Hills Oasis, Enderby Land. The isolated meltwater bacteria were identified as Proteobacteria, Actinobacteria, Firmicutes, and Bacteroidetes, where Proteobacteria and Actinobacteria were predominant. The isolated bacteria were able to grow in a relatively wide temperature range between 4 °C and 37 °C, with an optimal temperature range of 18–25 °C. Further, most of the isolates showed an ability to secrete lipases and proteases, and several of them were pigmented. Bacterial isolates from the genera *Pseudomonas* and *Acinetobacter* exhibited multi-resistance against β-lactams, sulfonamide, macrolide, diaminopyrimidines, and chloramphenicol antibiotics. This study shows that bacterial communities from the temporary meltwater ponds in East Antarctica consist of metabolically versatile bacteria that might be defined by their location near the sea and the close presence of animals, penguins and skuas in particular.

**Abstract:**

In this study, for the first time, we report the identification and characterization of culturable fast-growing bacteria isolated from the sea-affected temporary meltwater ponds (MPs) in the East Antarctica area of the Vecherny region (−67.656317, 46.175058) of the Thala Hills Oasis, Enderby Land. Water samples from the studied MPs showed alkaline pH (from 8.0 to 10.1) and highly varied total dissolved solids (86–94,000 mg/L). In total, twenty-nine bacterial isolates were retrieved from the studied MPs. The phylogenetic analysis based on 16S rRNA gene sequence similarities showed that the isolated bacteria belong to the phyla Proteobacteria, Actinobacteria, Firmicutes, and Bacteroidetes and the twelve genera *Pseudomonas*, *Shewanella*, *Acinetobacter*, *Sporosarcina*, *Facklamia*, *Carnobacterium*, *Arthrobacter*, *Brachybacterium*, *Micrococcus*, *Agrococcus*, *Leifsonia*, and *Flavobacterium*. Most of the isolated bacteria were psychrotrophs and showed the production of one or more extracellular enzymes. Lipolytic and proteolytic activities were more prevalent among the isolates. Five isolates from the Actinobacteria phylum and one isolate from the Bacteroidetes phylum had strong pigmentation. Antibiotic susceptibility testing revealed that most of the isolates are resistant to at least one antibiotic, and seven isolates showed multi-resistance.

## 1. Introduction

Due to the weather peculiarities, the polar region is considered to be one of the most versatile environments. During the summer period in polar areas, there are numerous temporary meltwater ponds (MPs) appearing as a result of snow and ice melting [1]. MPs are formed on the rough terrain, often between rocky ridges where there are conditions for the accumulation of the melted snow and snow-glacial water [2]. MPs sizes varies significantly depending on the location, presence of slopes, etc., and they can be flowing, low-flowing, or non-flowing and characterized by different physical parameters and chemical compositions providing individually distinct geochemical environments [3]. MPs are very dynamic and strongly connected with snow and ice since they appear for a short period due to snow and ice melting during the Antarctic summer (October–February). Temporary water reservoirs located on Antarctica’s shore are constantly exposed to various environmental abiotic (temperature fluctuations, freeze-thaw cycles, nutrient deficiency, abrupt chemical gradients, and increased salinity in habitats), biotic (plants and animals), and anthropogenic factors, and therefore, they usually have microbiota consisting of diverse cold-adapted organisms [4,5,6].

According to the authors’ knowledge, the microbial communities from Thala Hills oasis MPs are the least studied biotopes, and only MPs in the regions of the McMurdo Ice Shelf (MIS) and Dry Valleys have been described to some certain extent, with the main focus on the geochemistry of the water column [7,8,9], the microbial diversity of microbial mats [10] and sediments [11], surface water samples [12], and the water column [13]. Little attention has been given to the isolation and characterization of bacteria inhabiting MPs in Thala Hills oasis. Only a few recent studies that focused on the terrestrial biology, geology, and diversity of eukaryotes and prokaryotes in the snow and soil from this region are available [14,15,16,17]. Moreover, recently the characterization and biopotential of yeast cultures isolated from soil samples of East Antarctica were evaluated [18]. Due to the fact that bacteria in Antarctic meltwater ponds are important for driving biogeochemical cycles, sustaining essential chemical processes, and participating in a carbon sink and because they are able to tolerate environmental fluctuations, studying them could be especially useful for understanding environmental and ecological changes in Antarctica as well as discovering new microbial cell factories for modern biotechnology and biorefinery applications. For example, biotechnologically attractive enzymes (e.g., proteases, lipases, amylases, ureases, nucleases, β-galactosidases, and keratinases) [14,19,20,21,22,23,24,25] and pigments (e.g., carotenoids) [19,26,27,28,29,30,31] can be synthesized by culturable psychrophilic and psychrotrophic bacteria [32]. 

Approaches for investigating microbiota from a given ecosystem can be either culture-dependent or culture-independent. Despite the advantages of the culture-independent method, which provides more informative data on the diversity of microbial communities, the culture-dependent method, which leads to the isolation of culturable bacteria, remains a part of a traditional bioprospecting strategy in biotechnology [19]. In the present study, the culture-dependent methods were used.

Antibiotic resistance is a natural defense mechanism in bacteria [33], but it can be exacerbated due to the transportation of antibiotics to Antarctic areas by atmospheric and oceanic currents from outside the Antarctic region [33]. In addition, increasing the selective pressure of the anthropogenic and animal activity in Antarctica results in the spread of resistant strains [34], which may lead to the appearance of multi-resistant strains [35]. In order to understand to what extent meltwater bacteria possess antibiotic resistance, we performed antibiotic susceptibility testing for a total of twenty-three antibiotics and antimicrobial agents. 

The main aim of this study was to identify and characterize the fast-growing bacteria isolated from MPs in the Vecherny region of the Thala Hills Oasis, Enderby Land, which has limited anthropic pressure and is affected by the sea and the presence of animals, penguins and skuas in particular. To the authors’ knowledge, this is the first study reporting the identification and characterization of the culturable fast-growing bacteria isolated from the temporary meltwater ponds of Enderby Land, East Antarctica.

## 2. Materials and Methods

### 2.1. Sampling Sites and Sampling Procedure

Water samples were collected during the 5th Belarusian Antarctic Expedition in the austral summer season (January 2013) from the middle part of the water column of nine non-flowing MPs located in rock baths (0.1–3 m diameter, 2–100 m distance from the shoreline, 0.1–0.5 m deep, and 0–10 m above sea level). The key sampling site was located 800 m from the Belarussian Antarctic Station “Vechernyaya”, 14 km from the Russian Antarctic Station Molodezhnaya, and 2.7 km from the Adelie penguin colony at the Azure Cape (−67.656317, 46.175058) of the Vecherny region of the Thala Hills oasis in the central part of Enderby Land (East Antarctica) (Figure 1).

For the isolation of the fast-growing bacteria, water samples were collected in 20 mL sterile polypropylene tubes and kept at 4 °C before the isolation. For bacterioplankton analysis, water samples were collected in 100 mL sterile polythene bottles. Then, formalin, until a final concentration of 4%, was added to each sample for preservation. The collected samples were stored at 4 °C. For the physicochemical analysis, samples were collected in 100 mL non-sterile polythene bottles. All water samples were collected in two replicates on the same day.

### 2.2. Bacterioplankton and Physicochemical Analysis of Water Samples

Physicochemical parameters such as pH, total dissolved solids (TDS), and temperature were measured in situ at different time points (Day 1—11 January 2013, Day 2—14 January 2013, Day 3—17 January 2013, and Day 4—26 January 2013) with portable Combo pH/Conductivity/TDS Testers (HI98129 Low Range, HI98130 High Range) (Hanna Instruments, Kungsbacka, Sweden). 

For the evaluation of the quantitative parameters of bacterioplankton, samples collected on the first day (Day 1—11 January 2013) were used. The bacterial cell number was determined using the acridine orange method according to Hobbie et al. [36], and the measurements were taken with an Axiovert 25 epifluorescence inverted microscope (Carl Zeiss, Berlin, Germany ) equipped with a Nuclepore filter with a pore diameter of 0.2 μm and an AxioCam MRc camera (Carl Zeiss, Berlin, Germany). Water samples were examined under the 100× immersion lens. Pictures for each water sample were recorded in Carl Zeiss AxioVision Rel. 4.4 software (Carl Zeiss, Berlin, Germany) (10 in parallel from each filter (data not shown)). The processing of the obtained imaging data was conducted in the Image-Pro Plus program (Media Cybernetics, Rockville, MD, USA).

The correction of color and tone for the fluorescence images and the counting of objects (bacterioplankton) with the output of their geometric characteristics was performed by automated processing using an in-house algorithm created in the built-in Image-Pro Plus macro language. After the preliminary counting, the algorithm makes it possible to manually correct the counted objects (the separation of merged objects the removal of image artifacts from the counted objects, and the removal of wrongly identified objects). The estimation of the bacterioplankton was carried out using the corrected data. For the estimation of the bacterial cell number (BN), the following formula was used: BN = S × 10^6^ × a/s × V × 10, where S is the filter area, mm; 10^6^ is the recalculation of mm in µm; a is the sum of counted cells; s is the grid area, µm; V is the volume of the filtered sample; and 10 is a number of fields of view. The biomass was calculated according to the size of each bacterial cell. An in-house algorithm in Microsoft Excel was used to automatically enter the conversion formula in a spreadsheet and form an array of processing results.

### 2.3. Isolation of the Fast-Growing Bacteria

The isolation of the fast-growing bacteria was performed by the spread plating of 0.1 mL of each water sample in triplicates on meat peptone agar (MPA) (5.0 g/L peptone, 1.5 g/L meat extract, 1.5 g/L yeast extract, and 20.0 g/L agar, pH 7.0 ± 0.2) and cultivating for 14 days at 5 °C and 18 °C for isolating psychrophilic and psychrotolerant bacteria, respectively. Bacterial colonies that differed in phenotypic traits (the form of colonies, growth rate, and pigmentation) were selected for further purification. The growth rate was determined visually by daily observation, and we selected the colonies appearing at various time points during the first 2–6 days of the cultivation. The obtained single colonies were transferred onto new Petri dishes with MPA to obtain pure cultures. Isolated pure cultures were preserved in the following way: (1) the cultures were grown on the slanting full-fledged MPA agar in tubes for 6 days at 18 °C; (2) the obtained cells were washed and suspended in a mixture of meat peptone broth (MPB) (5.0 g/L peptone, 1.5 g/L meat extract, 1.5 g/L yeast extract, and 5.0 g/L NaCl with a final pH of 7.0 ± 0.2) and glycerol (20% of the final volume) with the ratio 1:1; (3) the suspended cells were stored at −80 °C.

### 2.4. Total DNA Extraction, Amplification, and Sequencing

For 16S rDNA sequencing, bacteria were cultivated on MPA agar for 7 to 10 days at 18 °C. Bacterial DNA was extracted using a DNA Preparation Kit PP-206 (Jena Biosciences, Jena, Germany). The extracted and purified DNA was stored at −20 °C. The fragment of 16S rDNA was amplified using universal bacterial primers 8f (5′-AGAGTTTGATCCTGGCTCAG-3′) and 1492r (5′-GGTTACCTTGTTACGACTT-3′) (Primetech, Minsk, Belarus), described previously [37]. Each 25 μL of the reaction mixture contained 10 μL of 2.5 × Flash buffer (ArtBioTech, Minsk, Belarus), 0.2 µL of each primer with the final concentration of 0.4 mM, 1 µL (≈10 ng) of bacterial DNA matrix, 0.25 μL of Flash polymerase (high-performance Pfu-polymerase, 2 U/μL) (ArtBioTech, Minsk, Belarus), and H_2_O (deionized) up to 25 μL. For the negative control, deionized water was added in an equivalent amount instead of the DNA matrix. Polymerase chain reaction (PCR) was performed in a SureCycler 8800 thermocycler (Agilent Technologies, Santa Clara, CA, USA). The initial denaturation was performed at 98 °C for 3 min, and 30 cycles of amplification consisted of denaturation at 98 °C for 30 s, annealing at 51 °C for 30 s, and elongation at 72 °C for 1 min, and the final extension phase was performed at 72 °C for 4 min.

The sequencing of 16S rRNA genes was carried out according to the Sanger method [38], and it was performed using the DNA Cycle Sequencing Kit PCR-401S (Jena Bioscience, Jena, Germany) and the following primers (Primetech, Minsk, Belarus): 926R-seq (5′-CCGTCAATTCATTTGAGTTT-3′), 336F-seq (5′-ACGGYCCAGACTCCTACG-3′), 522R-seq (5′-TATTACCGCGGCTGCTGGCAC-3′), and 918F-seq (5′-ACTCAAAKGAATTGACGGG-3′). All reactions were run according to the manufacturer’s protocols. The products of the sequencing reaction were detected using the automatic sequencer «4300 DNA Analyzer» (LI-COR Biosciences, Lincoln, NE, USA).

### 2.5. Analysis of 16S rRNA Sequencing Data

The obtained 16S rRNA data were preprocessed by editing and rendering in FASTA format using the e-Seq™ software V. 3.1.10 (LI-COR Biosciences, Lincoln, NE, USA). To obtain consensus sequences, all sequences for each isolate were aligned using AlignIR 2.1 (LI-COR Biosciences, Lincoln, NE, USA). The obtained sequences were compared to those available in the EzBioCloud database (ChunLab Inc., Seoul, Korea) [39] to choose reference sequences for the phylogenetic analyses and find similarities with the known strains. The phylogenetic tree was reconstructed using the MEGA 11 program [40]. The CLUSTAL W algorithm was used to align the sequences with the most similar orthologous sequences from the EzBioCloud database. The phylogenetic distance tree was inferred using the neighbor-joining analysis (NJ, *p*-distance matrix). The optimal mathematical model of nucleotide substitutions with the lowest Bayesian information criterion score was selected for the construction of the phylogenetic tree. The evolutionary history was inferred using the maximum likelihood method and the Tamura–Nei model [41]. To evaluate the confidence limits of the branching, a bootstrap analysis was performed on a 1000 replicate data set [42]. Bootstrap values greater than 70% of confidence are shown at the branching points. The strain archaea *Methanosarcina barkeri* Schnellen 1947 was chosen as a root for the phylogenetic tree. The obtained 16S rRNA sequences are deposited in the GenBank nucleotide sequence database (National Center for Biotechnology Information, Bethesda, MD, USA) under the accession numbers ON248060-ON248088.

### 2.6. Thermotolerance and Enzymatic Activity

To study thermotolerance, the bacterial isolates were grown on BHI (brain heart infusion) agar (Sigma Aldrich, St. Louis, MI, USA) at 4 °C, 10 °C, 18 °C, 25 °C, 30 °C, and 37 °C for up to 10 days with daily visual inspections of the cultures.

The production of the extracellular enzymes was evaluated by applying various plate-based assays using specific substrates—solid basal media containing: (1) for lipolytic activity, 10 g/L peptone, 5 g/L NaCl, 0.1 g/L CaCl_2_·2H_2_O, g/L 20 agar, and 10 mL (*v/v*) of Tween 80, pH 7.4; (2) for amylolytic activity, 10 g/L peptone, 5 g/L KH_2_PO_4_, 20 g/L agar, and 0.2% (*w/v*) soluble starch; (3) for protease activity, 6 g/L NaCl, 1.3% (*w/v*) nutrient broth, and 15% gelatin or calcium–casein agar (Condalab, Torrejón de Ardoz, Spain); (4) for DNase activity, DNase test agar (Condalab, Torrejón de Ardoz, Spain); (5) for urease activity, 1 g/L peptone, 5 g/L NaCl, 1 g/L glucose, 2 g/L KH_2_PO_4_, 0.012 g/L phenol red, 20 g/L agar, and 20 g/L urea, pH 6.8 +/− 0.2 (at 25 °C); (6) for β-galactosidase (β-GAL) activity, Luria broth agar with X-Gal (5-bromo-4-chloro-3-indolyl-β- D-galactopyranoside) and IPTG (isopropyl β- D-1-thiogalactopyranoside); and (7) for keratinase activity, 15 g/L chicken feather meal powder, 0.5 g/L NaCl, 0.3 g/L K_2_HPO, 0.4 g/L KH_2_PO_4_, and 15 g/L agar, pH 7.2. All plate-based assays were performed in duplicates at 18 °C for up to 10 days. The enzymatic activity was evaluated by estimating the colony growth, the formation of enzyme-specific zones, and/or the presence of media changes: (1) white precipitation zones of calcium salts around colonies for lipolytic activity; (2) clearance zones after flooding with iodine solution for amylolytic activity; (3) clearance zones for protease activity; (4) colorful zones for DNase activity; (5) the intensity of the pink color of the medium for urease activity; (6) the intensity of blue-colored colonies for β-galactosidase (β-GAL) activity; (7) the presence of a hydrolysis halo around the colony for keratinase activity. Catalase activity was determined by using the slide-drop method, where immediate bubble formation was observed after mixing a small amount of bacterial biomass with 3% H_2_O_2_.

### 2.7. Antibiotic Susceptibility Testing

Antibiotic susceptibility was evaluated by the Kirby–Bauer disc diffusion method [43] using Mueller–Hinton agar (Merck, Darmstadt, Germany), and it was performed in duplicates. The isolate *Carnobacterium iners* TMP 28 was excluded from the experiment due to weak growth. Based on the predominance of psychrotrophic bacteria and their optimal growth temperature in the range of 18–25 °C, two temperatures, 18 °C and 25 °C, were selected for the test. Susceptibility was evaluated for 18 antibiotics (3 of which were presented in two different concentrations) and 5 antibacterial agents. The following commercial disks for susceptibility testing (Bio-Rad, Hercules, CA, USA) were used: glycopeptide antibiotic (5 μg of vancomycin), β-lactam antibiotics (10 μg of ampicillin, 20 μg of amoxicillin, 10 μg of clavulanic acid, 30 μg of cefuroxime, 10 μg of imipenem, and 30 μg of ceftriaxone), quinolones (5 μg of ciprofloxacin and 30 μg of nalidixic acid), diaminopyrimidines (5 μg of trimethoprim) and diaminopyrimidine coupled with sulfonamide (1.25 μg of trimethoprim and 23.75 μg of sulfamethoxazole), rifamycin antibiotics (5/30 μg of rifampicin), macrolide antibiotic (15 μg of erythromycin, 15 μg of clarithromycin, and 15 μg of azitromycin), aminoglycoside antibiotic (10/120 μg of gentamicin, 10 μg of tobramycin, 10/300 μg of streptomycin, and 30 μg of kanamicyn), tetracycline antibiotics (30 μg of doxycycline and 30 μg of tetracycline), 30 μg of chloramphenicol, and 100 μg of nitrofurantoin. The strains *Escherichia coli* ATCC 25922, *Pseudomonas aeruginosa* ATCC 27853, and *Staphylococcus aureus* ATCC 29213 were used for the quality control of the antibiotic assays. The data analysis for the control cultures was performed according to the breakpoints established by the EUCAST [44] and CLSI [45] documents. The data analysis for the Antarctic bacterial isolates was performed as described by Daniela et al. [35]. The inhibition zones ≤ 15 mm in diameter (including the disc) were considered as a breakpoint to define the resistance, and zones ≤ 20 mm were defined as intermediate. 

## 3. Results

### 3.1. Physicochemical Characterization of Water Samples from Meltwater Ponds

Size, depth, and physicochemical parameters such as pH, temperature (T), and total dissolved solids (TDS) were measured for all MPs on different days to follow changes over time. The pH values of all water samples were alkaline (from 8.0 to 10.1). TDS varied considerably from 86 mg/L to 94,000 mg/L. An increases in TDS for all samples were recorded over time because of the evaporation process. The temperature of the water samples was between 12 °C and 17.5 °C (Table 1). All ponds were characterized by the absence of water flow.

The quantitative characteristics of bacterioplankton varied from 0.95 ± 0.12 × 10^6^ to 4.52 ± 0.67 × 10^6^ cells/mL, and the biomass was in a range from 0.151 ± 0.034 to 0.939 ± 0.280 mg/L (Table 1). For all ponds, a predominance of orange or brown benthos was visually observed. The presence of clean water in the upper layer was observed for all ponds except for MP-1 and MP-7, which had considerable water blooms. The presence of organic clusters of biomass with a diameter of ≈0.5 cm was observed in MP-1 and MP-6. Ponds MP-1 and MP-2 were located 30 m from the skua nest and served as swimming places for skuas. For the pond MP-8, a white salt crust formation along the edges was observed (Figure 1). 

### 3.2. Isolation and Phylogenetic Characterization of the Meltwater Fast-Growing Bacteria

Due to the fact that the isolation of bacteria was conducted for two weeks and all selected isolates retrieved within this time were able to produce single colonies within 2–6 days, we concluded that these isolated bacteria could be considered to be fast-growing [14,46,47]. In total, twenty-nine fast-growing bacteria with different colony types (colony morphology, size, and pigmentation) and growth rates were isolated from nine MPs of East Antarctica. The highest numbers of bacterial isolates were obtained from the ponds MP-9 (7 isolates), MP-5 (5 isolates), and MP-6 (4 isolates), while other ponds were characterized by 1 to 3 bacterial isolates (Table 1). Five isolates showed strong pigmentation: *Flavobacterium degerlachei* TMP13, *Arthrobacter* sp. TMP15, *Agrococcus citreus* TMP23, and *Leifsonia* sp. TMP30 showed a strong yellow color, and isolate *Arthrobacter agilis* TMP24 had a salmon color (Figure 2). All isolates were deposited in the Belarusian Collection of Non-pathogenic Microorganisms at the Institute of Microbiology of the National Academy of Sciences of Belarus (Minsk, Belarus) (Table 2).

16S rRNA sequencing was used to study the phylogenetic relationships of the isolated fast-growing bacteria. A comparative 16S rRNA gene sequence analysis revealed the similarity of the isolate sequences to the EzBioCloud database sequences, with high similarity percentages from 98.47% to 100 % (Table 2, Figure 3). The isolated culturable fast-growing bacteria were phylogenetically related to four phyla, Proteobacteria, Actinobacteria, Firmicutes, and Bacteroidetes, and represented twelve genera. Proteobacteria was the first predominant phylum among the isolates (17 out of 29 isolates), and it was represented by three genera: *Pseudomonas* (12/17), *Shewanella* (4/17), and *Acinetobacter* (1/17). Actinobacteria was the second predominant phylum (6/29), and it was represented by five genera: *Arthrobacter* (2/6), *Brachybacterium* (1/6), *Micrococcus* (1/6), *Agrococcus* (1/6), and *Leifsonia* (1/6). The Firmicutes phylum (5/29) was represented by three genera: *Carnobacterium* (3/5), *Sporosarcina* (1/6), and *Facklamia* (1/6). The Bacteroidetes phylum was represented by only one isolate belonging to the genus *Flavobacterium*. Isolates from the Proteobacteria phylum were detected in all studied ponds except MP-8, while isolates of the other phyla were detected only in ponds MP-4–MP-9 (Table 2).

Isolates belonging to the genera *Pseudomonas*, *Carnobacterium*, and *Arthrobacter* were represented by several species such as *P. lundensis*, *P. peli*, *P. leptonychotis*, *C. funditum*, *C. iners*, *C. inhibens*, *Arthrobacter* sp., and *Ar. agilis*. Isolates from the genera *Shewanella*, *Acinetobacter*, *Sporosarcina*, *Facklamia*, *Flavobacterium*, *Brachybacterium*, *Microccocus*, *Agrococcus*, and *Leifsonia* were represented by the species *S. baltica*, *Ac. lwoffii*, *S. globispora*, *F. tabacinasalis*, *F. degerlachei*, *B. paraconglomeratum*, *M. luteus*, *Ag. citreus*, and *Leifsonia* sp., respectively. Due to the low resolution of 16S rRNA gene sequence analysis, the species affiliation for some isolates of the genera *Pseudomonas*, *Arthrobacter*, and *Leifsonia* could not be determined (Table 2). 

### 3.3. Thermotolerance and Enzymatic Activity

To evaluate the thermotolerance and identify the optimal growth temperature, bacterial isolates were cultivated at six temperatures (4 °C, 10 °C, 18 °C, 25 °C, 30 °C, and 37 °C). The majority of the isolates were found to be psychrotolerant, with an optimal growth temperature of 18 °C, but were able to grow at 4 °C to 25 °C, 30 °C, and 37 °C. All bacterial isolates (28/29) except *Agrococcus citreus* TMP23 grew at 10 °C and 18 °C, and most of the strains (23/29) were able to grow at 4 °C. Four isolates, *Pseudomonas lundensis* TMP2, TMP3, and TMP4 and isolate *Acinetobacter lwoffii* TMP6, were able to grow at all tested temperatures. Bacterial isolates *Facklamia tabacinasalis* TMP29, *Brachybacterium paraconglomeratum* TMP16, and *Microccocus luteus* TMP21 did not grow at 4 °C but were able to grow at 30 °C or 37 °C. Two isolates, *Carnobacterium funditum* TMP27 and *Carnobacterium iners* TMP28, were able to grow at 4 °C but not at 25 °C and could be considered psychrophiles (Table 2).

Most of the isolated fast-growing meltwater bacteria showed one or more enzymatic activities, except the isolates *Carnobacterium funditum* TMP27, *Carnobacterium iners* TMP28, and *Facklamia tabacinasalis* MP29, which showed slow growth on minimal selective media. A high level of enzymatic activity (three or more enzymes) was detected for *Brachybacterium paraconglomeratum* TMP16 and *Microccocus luteus* TMP21 and for all isolates of *Pseudomonas lundensis* and *Shewanella baltica*. Lipolytic activity was the most common enzymatic activity among the isolates, where all strains of *Shewanella baltica*, *Pseudomonas leptonychotis*, *Pseudomonas peli*, and *Acinetobacter lwoffii* TMP6, *Carnobacterium inhibens* TMP12, *Arthrobacter agilis* TMP24, and *Micrococcus luteus* TMP21 showed an ability to hydrolyze Tween 80. Proteolytic activity was detected for twelve isolates. The degradation of both casein and gelatin were detected for all isolates of *Pseudomonas lundensis* and *Micrococcus luteus* TMP21. The utilization of gelatin was observed for all isolates of *Shewanella baltica*, while casein was degraded by all isolates of *Pseudomonas leptonychotis* and *Brachybacterium paraconglomeratum* TMP16. Five isolates, *Pseudomonas lundensis* TMP2, TMP3, TMP4, *Sporosarcina globispora* TMP10, and *Brachybacterium paraconglomeratum* TMP16, showed urease activity. Amylase activity was detected only for two isolates, *Brachybacterium paraconglomeratum* TMP16 and *Micrococcus luteus* TMP21. The production of deoxyribonuclease was detected for all isolates of *Shewanella baltica*. β-galactosidase activity was detected only for *Brachybacterium paraconglomeratum* TMP16. Catalase activity was determined for all isolates from the phyla Bacteroidetes and Actinobacteria and for several Proteobacteria isolates. The production of keratinase was not detected for the isolated bacteria (Table 2).

### 3.4. Antibiotic Susceptibility 

In total, twenty-eight isolated meltwater bacteria were tested for susceptibility towards twenty-three antibiotics and antibacterial agents, where twenty of them were broad-spectrum antibiotics, one was an antibiotic against Gram-positive bacteria (Vancomycin), one was an antibacterial agent active against Gram-negative bacteria (Nalidixic acid), and one was an antibacterial against most Gram-positive cocci and *E. coli* (Nitrofurantoin). Rifampicin, gentamicin, and streptomycin were presented in two concentrations (Figure 4). Due to the absence of growth on Mueller–Hinton agar at 25 °C, the susceptibility screening for isolates *Flavobacterium degerlachei* TMP13, *Carnobacterium funditum* TMP27, and *Leifsonia* sp. TMP30 was conducted only at 18 °C. We observed that among the tested bacteria many were not susceptible to nitrofurantoin (24 and 22 isolates at 18 and 25 °C, respectively), and all Gram-negative and Gram-positive bacteria were resistant to vancomycin and nalidixic acid, respectively, due to natural resistance. These data were not included in the calculation of the susceptibility profile of the analyzed bacteria. The studied bacteria showed higher susceptibility towards the evaluated antibiotics at 25 °C than at 18 °C. For some isolates, temperature-induced changes in antibiotic susceptibility were observed (Figure 4). 

Twenty-five isolates were resistant to at least one antibiotic, and seven isolates showed different levels of multi-resistance. Many isolates (>10) were resistant to ampicillin, cefuroxime, amoxicillin-clavulanic acid, and trimethoprim. Very few isolates (<3) were resistant to imipenem, ciprofloxacin, 30 μg of rifampicin, 120 μg of gentamicin, 300 μg of streptomycin, and doxycycline. Variation in the resistance was detected towards antibiotics present in different concentrations (gentamicin, streptomycin, and rifampicin), and resistance was lower for higher concentrations. None of the isolates were resistant to a high concentration of streptomycin (Figure 4).

Most of the Gram-negative isolates were resistant to β-lactam-type antibiotics such as ampicillin, amoxicillin-clavulanic acid, cefuroxime, and trimethoprim. Among the Gram-negative isolates, *Acinetobacter lwoffii* TMP6 and all *Pseudomonas lundensis* isolates showed multiple antibiotic resistance to more than ten antibiotics from the classes β-lactams, sulfonamide, macrolide, and chloramphenicol. Among the Gram-positive isolates, the highest level of resistance was detected against aminoglycoside antibiotics (tobramycin, kanamycin, and low concentrations of streptomycin) and trimethoprim. Isolates *Brachybacterium paraconglomeratum* TMP16, *Agrococcus citreus* TMP23, *Carnobacterium inhibens* TMP12, and *Facklamia tabacinasalis* TMP29 were resistant to five and more antibiotics, mostly aminoglycosides and trimethoprim. Isolates *Pseudomonas peli* TMP9 and TMP25 and *Sporosarcina* sp. TMP10 were susceptible to all tested antibiotics (Figure 4).

## 4. Discussion

In the framework of this study, we, for the first time, characterized bacteria isolated from the sea-affected terrestrial temporary non-flowing meltwater ponds in the Thala Hills oasis (Enderby Land, East Antarctica), which are small water reservoirs formed on the unevenness of the terrain due to snow and ice melting. 

A physicochemical characterization of the water samples showed that the water in these MPs has a high alkaline pH that can possibly be explained by the photosynthetic activity of cyanobacteria [48]. We observed high variability in TDS, the number of bacteria, and the productivity of bacterioplankton between different ponds, which can be due to the absence of water flow, small size (diameter 0.1–3 m), low depth (0.1–0.5 m), chemical composition of rock-forming minerals, water evaporation process, and effect of sea aerosols due to the location along the shoreline. A similar variability in TDS was reported for meltwater ponds from other Antarctic regions [7,49]. A previous study of the temporary meltwater ponds of the same area in East Antarctica [3] showed that the maximum number of bacterial cells and microbial biomass in non-flowing ponds (which we study in this paper) are higher than in flowing and weakly flowing ponds and were closer to the number measured for the green snow sample taken in the same area [14]. Thus, when calculating the biomass of bacterioplankton per 1 L, it can be concluded that the productivity of bacterioplankton in stagnant water ponds is the highest, more than 20 times higher than the productivity of bacterioplankton in flowing water ponds, and comparable to the productivity of surface snow samples. 

Physicochemical parameters such as salinity and pH can have significant direct and indirect impacts on the formation of microbial communities as well as their diversity and ecological responses [50]. Combining culture-based and non-culture-based approaches is desirable for describing novel microbial communities, but in this study, one of the aims was to isolate culturable bacteria that could be further used in bioprospecting for the production of different valuable products. Therefore, the culture-based approach was applied. For example, the production of pigments such as carotenoids is common among Antarctic bacteria because they play a key role in the adaptation to the cold environment through the modulation of membrane fluidity and protect bacterial cells against ultraviolet radiation [26,27,28,30,31]. Thus, at a very first step, in order to isolate bacterial isolates from different taxonomic groups, bacterial colonies with considerably different phenotypic traits (shape and size of colonies, growth rate, pigmentation) were selected, resulting in twenty-nine fast-growing bacterial isolates.

16S rRNA sequencing indicated the presence of similar types of bacteria as reported for other meltwater ponds and environments in Antarctica. The isolated meltwater bacteria were identified as Proteobacteria, Actinobacteria, Firmicutes, and Bacteroidetes, in accordance with the previously reported studies [23,51,52]. Among the isolated bacteria, Proteobacteria (with the predominance of the genus *Pseudamonas*) were the dominating group, with seventeen isolates that were isolated from all meltwater ponds except MP- 8. Proteobacteria bacteria may dominate in culture-based studies due to their ability to grow rapidly on nutrient-rich media such as MPA and effectively compete in heterotrophic conditions [23,53]. Proteobacteria are often identified as the dominating member of Antarctic microbial communities [11,51,52]. The predominance of Proteobacteria and their presence in almost all studied ponds may have a connection to the alkaline pH and small depth of the ponds since it was previously reported that Proteobacteria are the dominant phylum in the samples taken from the bottom of water bodies, while Bacteroidetes are predominant in surface water samples [13]. The second most abundant phylum was Actinobacteria, with genus *Arthrobacter* dominating. The isolation of *Arthrobacter* and *Leifsonia* was shown for green snow samples from Antarctica [14]. Such genera as *Agrococcus*, *Arthrobacter*, *Brachybacterium*, and *Micrococcus* were found in an association with marine macroalgae from Antarctica [54], and they are often retrieved from different Antarctic samples using culture-based techniques [55]. The presence of Actinobacteria members may lead to their co-existence with marine macroalgae and high adaptability since they can tolerate a wide range of environmental gradients, such as temperature, pH, and salinity [56]. The production of pigments by Actinobacteria may be promising for biotechnological applications [31,54]. The genera *Carnobacterium* and *Sporosarcina* were predominant from the Firmicutes phylum and were previously isolated from other Antarctic environments (lakes, ponds, and permafrost ice) [23]. Retrieving Firmicutes bacteria could be explained by their ability to form endospores, which are extremely resistant to adverse environments [57]. Bacteroidetes were represented by only one isolate, which may be due to the low selectivity of the medium used and the overall predominance of Proteobacteria, as reported by Carmen et. al. [23]. Our results indicate that the isolated microbial communities from MPs vary, and a similar observation was shown before for the microbial communities of water bodies from the Ross Sea region, where the abundance of Proteobacteria, Bacteroidetes, and Actinobacteria within different ponds was highly variable between years, with a link to climate-driven factors [12].

Interestingly, that bacterial species isolated from MPs in this study were previously found in both cold and non-cold environments, while most of the isolated bacterial species were similar to those found in other types of samples from cold polar and non-polar regions. For example, other researchers isolated *Shewanella baltica* [58], *Pseudomonas lundensis* [59,60], *Pseudomonas leptonychotis* [61], *Carnobacterium funditum* [62], *Carnobacterium iners*, *Flavobacterium degerlachei* [63], *Arthrobacter agilis* [64], *Micrococcus luteus* [21], and *Leifsonia rubra* [14,65] from water, soil, and animal samples of polar origin. The species *Pseudomonas lundensis*, *Pseudomonas peli* [25], *Acinetobacter lwoffii* [66], *Sporosarcina globispora* [67], *Carnobacterium inhibens* [68], *Arthrobacter alpinus* [69,70], and *Arthrobacter agilis* were isolated from cold non-polar environments such as frozen food, non-polar permafrost sediments, glaciers, glacial currents, a subglacial lake in the Himalayas, and Siberian permafrost. Since in this study we isolated bacteria that were previously found in cold polar and non-polar regions, it could be hypothesized that these species have especially flexible metabolic machinery that allows them to adapt and survive in distinct cold regions [51].

Screening for thermotolerance showed that many of the isolated bacteria are psychrotrophs and can grow within a wide temperature range from 4 up to 25, 30, or 37 °C, and only two isolates, *C. funditum* TMP27 and *C. iners* TMP28, were classified as psychrophilic since they could grow at 18 °C but not at 25 °C [29,71]. The appearance of psychrotrophs is common in cold environments, as they have well-developed metabolic responses and nutrient adaptability to the wide temperature fluctuation common for Antarctica, as previously reported [22,72,73,74].

The production of a wide range of enzymes by cold-adapted psychotropic and psychrophilic bacteria was reported before, and a similar observation was obtained in this study. Screening for enzymatic activity at the optimal growth temperature of 18 °C showed that most of the isolates (predominantly from the *Pseudamonas* genus) possess lipolytic activity that is in accordance with previous studies [19,75,76]. In oligotrophic Antarctic water ecosystems, characterized by low temperatures and low light intensity, phytoplankton stores up to 80% of available carbohydrates and other C-sources in the form of lipids, which leads to an increase in the production of lipase/esterase enzymes by the microbes present in these ecosystems [19,77]. Some isolates have shown urease activity that could be explained by the presence of seals close to the sampling sites or by the presence of other natural sources of urea, as described by Tara et. al. [78]. Urea could be an important source of nitrogen in polar systems [78]. Urease activity correlates with antibiotic multi-resistance, which is often explained by the presence of the ABC transporter system used for the transportation of urea inside cells [79] and antibiotics outside [34]. *Shewanella* isolates showed DNase activity, and this is in accordance with the previous results where it was shown that *Shewanella* can utilize DNA as a sole source of carbon, nitrogen, and phosphorus in media that is lacking these nutrients under aerobic and anaerobic conditions [80]. The production of catalase, protecting cells from the toxic stress-induced reactive oxygen species (ROS) [20], was detected for more than half of the isolates. Keratinolytic activity was not possessed by any of the studied isolates, while the production of keratinases by Antarctic bacteria was previously reported [81]. The production of extracellular enzymes such us lipases, proteases, urease, and DNases by isolates from MPs indicates that these microbes play an important role in the breakdown and subsequent mineralization of MP organic matter. Bacteria could contribute to the biotransformation of organic carbon, sulfur, and nitrogenous compounds and may play a key role in the food webs and nutrient cycling of the pond’s ecosystem [82,83].

As the studied MPs are (1) located in the shore area and affected by the sea, (2) characterized by average human activity (3–10 people per season), and (3) the presence of birds (penguins and skuas) and sea animal (seals), their microbial communities potentially could be affected by the transport of resistance genes. Therefore, we decided to evaluate the antibiotic susceptibility of the bacteria isolated from these MPs. We recorded a relatively high level of resistance among the tested isolates (more than six resistant isolates) towards β-lactams (ampicillin, amoxicillin-clavulanic acid, and cefuroxime), macrolide (erythromycin), aminoglycoside antibiotic (streptomycin (in low concentrations) and tobramycin), trimethoprim and trimethoprim with sulfamethoxazole. Similar results were reported for the bacterial isolates from other Antarctic environments [20,35,72,84,85]. The presence of migratory birds in the studied area could generate a selective pressure on the local microbiota and contribute to the spreading of antibiotic resistance [35]. Thus, one of the most resistant isolates was from the species *Pseudomonas lundensis*, showing a high level of resistance to thirteen antibiotics, was isolated from MP- 1 and MP-2, which was located 30 m from the skua nest and served as a swimming place for skuas. Other isolates with low levels of resistance were from meltwater pounds MP-3, MP-5, MP-6, MP-8, and MP-9, which did not serve as a swimming places for skuas. Isolates of *Acinetobacter lwoffii* TMP6 were resistant to fifteen antibiotics, *Brachybacterium paraconglomeratum* TMP16 and *Agrococcus citreus* TMP23 were resistant to nine antibiotics, and *Facklamia tabacinasalis* TMP29 and *Carnobacterium inhibens* TMP12 were resistant to six and seven antibiotics, respectively. Isolates from ponds MP-4 and MP-7 showed high levels of susceptibility towards the tested antibiotics. Its known that bacteria from the phyla Proteobacteria, Firmicutes, and Actinobacteria could encode multiple resistance mechanisms [34]. One of the most common resistance strategies in bacteria is multidrug efflux pumps. Multidrug efflux pumps have broad specificity and confer resistance to a wide range of antibiotics [34,84]. It was also shown that due to the presence of many antibiotic resistance genes Proteobacteria could have taxonomic dominance in microbial communities. Interestingly, in this study, the resistance of bacterial isolates from the phylum Proteobacteria varied a lot—from isolates with multi-resistance to isolates susceptible to all tested antibiotics. Summarizing our observations, the bacterial communities of MPs could be used as bioindicators in Antarctica to track antibiotic resistance gene mobilization to the polar regions and the impact of human or animal presence.

## 5. Conclusions

Based on the obtained results, we can hypothesize that the bacterial communities of the temporary meltwater ponds in East Antarctica include metabolically versatile bacteria, as among the isolated species, many are commonly found in both polar and non-polar cold regions. The isolated bacteria were shown to be psychrotrophic and were able to grow at a wide range of the temperatures from 4 to 37 °C, and the optimal temperature varied from 18 to 25 °C. It seems that the physiological properties, such as the enzymatic activity and antibiotic susceptibility, of the isolated meltwater bacteria are defined by the location of animals, penguins and skuas in particular, close to the MPs. The production of enzymes and pigments detected in this study may be promising for biotechnological applications. Antibiotic susceptibility testing revealed that most of the isolates were resistant to at least one antibiotic, and seven isolates showed multi-resistance from six to fifteen out of twenty-three tested antibiotics and antibacterial agents. In summary, isolates from Antarctic MPs may have biotechnological potential and could be used as bioindicators to track antibiotic resistance gene mobilization and the impact of human or animal presence in polar regions.

## Figures and Tables

**Figure 1 biology-11-01143-f001:**
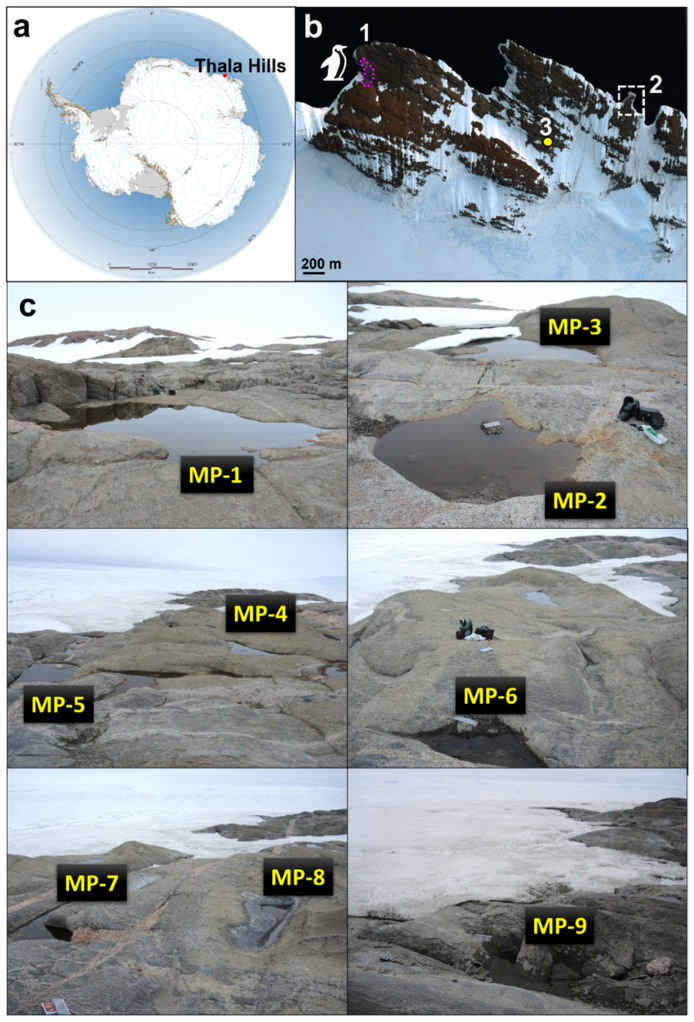
Sampling sites. (**a**) Location of the Thala Hills oasis in the coastal area of East Antarctica (marked by the red circle). (**b**) Satellite image of the eastern part of the Thala Hills oasis (1—Adelie penguin colony area, 2—sampling sites area, 3—location of the Belarussian Antarctic Station “Vechernyaya” in the eastern part of the Thala Hills oasis (marked by the yellow circle)). (**c**) Photos of the studied temporary meltwater ponds (MPs).

**Figure 2 biology-11-01143-f002:**
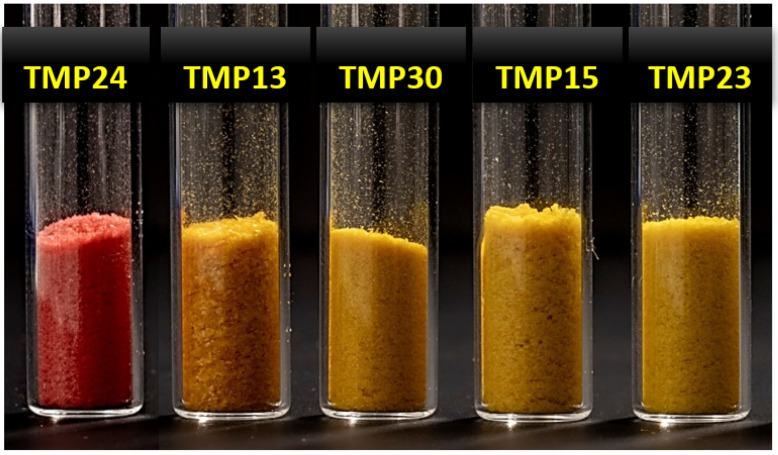
Freeze-dried biomass of pigmented isolates: *Flavobacterium degerlachei* TMP13, *Arthrobacter* sp. TMP15, *Agrococcus citreus* TMP23, *Arthrobacter agilis* TMP24, and *Leifsonia* sp. TMP30.

**Figure 3 biology-11-01143-f003:**
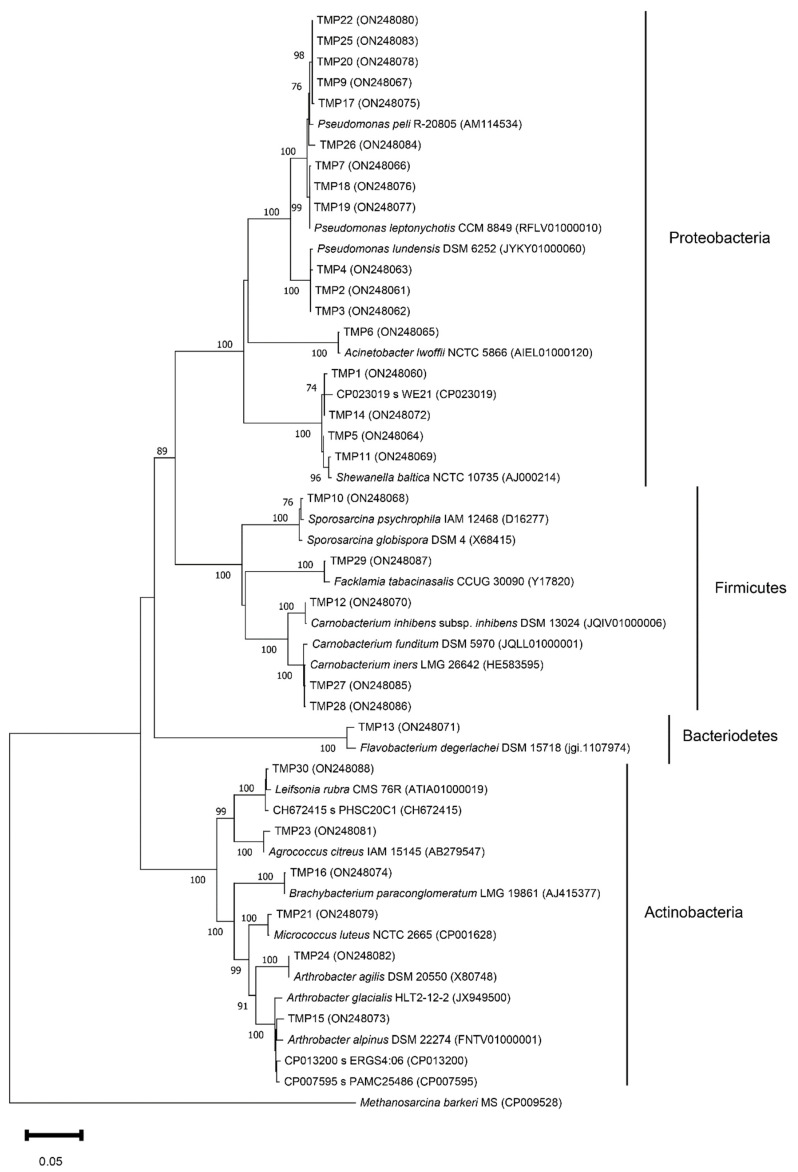
Phylogenetic tree based on 16S rDNA sequences of the MP isolates. Evolutionary history was inferred using the maximum likelihood method and the Tamura–Nei model. The node numbers represent the percentage of bootstrap replicates of 1000 resamplings (values below 70% are not shown). The scale bar represents substitutions per nucleotide position. All accession numbers are in parentheses following the bacterial strain.

**Figure 4 biology-11-01143-f004:**
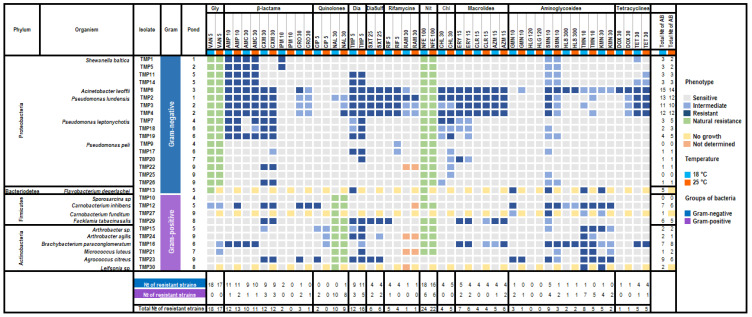
Heatmap of the antimicrobial susceptibility profiles of bacterial isolates from temporary meltwater ponds. Antibiotics and antibacterial agents (AB): Gly—Glycopeptide, Dia—Diaminopyrimidines, Sulf—Sulfanilamide, Nit—Nitrofurantoin, Chl—Chloramphenicol; VAN = 5 μg of vancomycin, AMP = 10 μg of ampicillin, AMC = 20 μg of amoxicillin + 10 μg of clavulanic acid, CXM = 30 μg of cefuroxime, IMP = 10 μg of imipenem, CRO = 30 μg of ceftriaxone, CIP = 5 μg of ciprofloxacin, NAL = 30 μg of nalidixic acid, TMP = 5 μg of trimethoprim, SXT = 1.25 μg of trimethoprim + 23.75 μg of sulfamethoxazole, RIF = 5 μg of rifampicin, RAM = 30 μg of rifampicin, ERY = 15 μg of erythromycin, CLR = 15 μg of clarithromycin, AXM = 15 μg of azithromycin, GMN = 10 μg of gentamicin, HGL = 120 μg of gentamicin, TMN = 10 μg of tobramycin, SMN = 10/300 μg of streptomycin, HLS = 300 μg of streptomycin, KMN = 30 μg of kanamycin, DOX = 30 μg of doxycycline, TET = 30 μg of tetracycline, CHL = 30 μg of chloramphenicol, NFE = 100 μg of nitrofurantoin. For each isolate, the total number of antibiotics to which it showed resistance was estimated separately for different temperatures (18 °C and 25 °C) and these values are shown in the second last two right columns. For each antibiotic, the total number of resistant strains was estimated separately for Gram-positive and Gram-negative bacteria and total number and shown in the last three rows.

**Table 1 biology-11-01143-t001:** Visual observation and physicochemical parameters measured in situ at different time points and morphometric parameters of bacterioplankton.

	Day	Temporary Meltwater Pond Number																	
		MP-1		MP-2		MP-3		MP-4		MP-5		MP-6		MP-7		MP-8		MP-9	
Physicochemical parameters																			
Size, m	Day 1 *	5 × 5		4 × 5		1.5 × 2		2 × 2		3 × 2		1 × 0.5		1.5 × 1		0.5 × 0.5		1 × 1	
Depth, m		0.3		0.3		0.2		0.25		0.25		0.3		0.5		0.1		0.2	
pH		10.1		9.8		8.0		10.0		9.6		9.9		9.5		8.3		10.0	
TDS *, mg/LT, °C		TDS	T	TDS	T	TDS	T	TDS	T	TDS	T	TDS	T	TDS	T	TDS	T	TDS	T
	Day 1	271	13	631	14	86	16	458	16	580	13	1902	13	4470	12	94,000	12	1640	13
	Day 2	ND	ND	859	18	122	17.5	585	17.3	730	14.4	PD *	PD	ND	ND	PD	PD	PD	PD
	Day 3	364	9.3	940	11.4	163	12.3	740	10.5	1006	8.9			5684	10.2				
	Day 4	401	ND	1209	ND	355	ND	1126	ND	790	ND			6385	ND				
Visual observation																			
WB	Day 1	+		−		−		−		−		−		+		−		−	
BC		B		O, B		O		O, G, B		O, G		B		O		Gr, B		−	
OC		+		−		−		−		−		+		−		−		−	
Skuas		+		+		−		−		−		−		−		−		−	
Salt		−		−		−		−		−		−		−		+		−	
Bacterioplankton parameters																			
BN	Day 1	4.52 ± 0.67		1.32 ± 0.17		1.52 ± 0.26		3.01 ± 0.36		0.95 ± 0.12		2.82 ± 0.41		2.22 ± 0.28		4.46 ± 0.40		2.02 ± 0.32	
BB		0.47 ± 0.17		0.16 ± 0.06		0.32 ± 0.12		0.27 ± 0.12		0.15 ± 0.03		0.50 ± 0.14		0.31 ± 0.11		0.94 ± 0.28		0.77 ± 0.20	
NBI		2		3		1		3		5		4		2		2		7	

* Day 1—11 January 2013, Day 2—14 January 2013, Day 3—17 January 2013, and Day 4—26 January 2013; T—temperature, TDS—total dissolved solids, PD—pond dried; ND—not determined; WB—Water bloom; OC—Organic matter clusters; Skuas—Skuas observed; Salts—Salt on the edges; BC—Benthos color: B—brown, O—orange, G—green, Gr—gray; “−”—not observed, “+”—observed; BN—bacterial cell number ×10^6^ cells/mL, BB—bacterial biomass, mg/L; NBI—number of bacterial isolates.

**Table 2 biology-11-01143-t002:** 16S rRNA gene sequence affiliation to the closest phylogenetic neighbors, isolation temperature, and physiological parameters of the bacteria isolated from Antarctic temporary meltwater ponds.

Isolate Code ^MP^ *	Collection Number	Gen Bank Accession Number	Nearest Taxonomic Neighbor by EzBioCloud Alignment	Identity (%)	Isolation Temperature (°C)	Thermotolerance (°C)	Enzymatic Activity at 18 °C
Proteobacteria							
TMP1 ^1^	BIM B-1565	ON248060	*Shewanella* *baltica* NCTC 10735	100	5	4–30	L *; Gel; D; C
TMP5 ^2^	BIM B-1557	ON248064	*Shewanella baltica* NCTC 10735	99.72	18	4–30	L; Gel; D; C
TMP11 ^5^	BIM B-1561	ON248069	*Shewanella baltica* NCTC 10735	99.66	5	4–30	L; Gel; D
TMP14 ^5^	BIM B-1563	ON248072	*Shewanella* WE21	99.38	5	4–30	L; Gel; D
			*Shewanella baltica* NCTC 10735	99.04			
TMP6 ^3^	BIM B-1558	ON248065	*Acinetobacter* *lwoffii* NCTC 5866	99.79	18	4–37	L; C
TMP2 ^1^	BIM B-1554	ON248061	*Pseudomonas lundensis* DSM 6252	99.86	18	4–37	Cas; Gel; U; C
TMP3 ^2^	BIM B-1555	ON248062	*Pseudomonas lundensis* DSM 6252	99.86	5	4–37	Cas; Gel; U; C
TMP4 ^2^	BIM B-1556	ON248063	*Pseudomonas lundensis* DSM 6252	99.79	18	4–37	Cas; Gel; U; C
TMP7 ^4^	BIM B-1559	ON248066	*Pseudomonas leptonychotis* CCM 8849	99.93	5	4–37	L; Cas
TMP18 ^6^	BIM B-1568	ON248076	*Pseudomonas leptonychotis* CCM 8849	100	18	4–30	L; Cas
TMP19 ^6^	BIM B-1566	ON248077	*Pseudomonas leptonychotis* CCM 8849	100	18	4–30	L; Cas
TMP9 ^4^	BIM B-1560	ON248067	*Pseudomonas peli* R-20805	99.52	5	4–25	L
TMP17 ^6^	BIM B-1569	ON248075	*Pseudomonas peli* R-20805	99.38	5	4–25	L
TMP20 ^7^	BIM B-1546	ON248078	*Pseudomonas peli* R-20805	99.52	5	4–25	L
TMP22 ^9^	BIM B-1552	ON248080	*Pseudomonas peli* R-20805	99.52	5	4–25	L
TMP25 ^9^	BIM B-1542	ON248083	*Pseudomonas peli* R-20805	99.52	18	4–25	L
TMP26 ^9^	BIM B-1548	ON248084	*Pseudomonas peli* R-20805	99.11	5	4–25	L
Bacteroidetes							
TMP13 ^5^	BIM B-1562	ON248071	*Flavobacterium* *degerlachei* DSM 15718	98.47	18	4–25	C
Firmicutes							
TMP10 ^4^	BIM B-1539	ON248068	*Sporosarcina* *globispora* DSM 4	99.59	18	4–30	U; C
			*Sporosarcina psychrophila* IAM 12468	99.59			
TMP12 ^5^	BIM B-1540	ON248070	*Carnobacterium inhibens* subsp. *inhibens* DSM 13024	100	5	4–30	L
TMP27 ^9^	BIM B-1541	ON248085	*Carnobacterium* *funditum* DSM 5970	100	18	4–18	ND
TMP28 ^9^	BIM B-1544	ON248086	*Carnobacterium iners* LMG 26642	99.86	18	4–18	ND
TMP29 ^8^	BIM B-1577	ON248087	*Facklamia* *tabacinasalis* CCUG 30090	99.46	18	10–30	ND
Actinobacteria							
TMP15 ^5^	BIM B-1549	ON248073	*Arthrobacter* ERGS4:06	98.97	5	4–25	C
			*Arthrobacter* PAMC25486	98.90			
			*Arthrobacter alpinus* DSM 22274	98.70			
			*Arthrobacter glacialis* HLT2-12-2	98.70			
TMP24 ^9^	BIM B-1543	ON248082	*Arthrobacter* *agilis* DSM 20550	100	18	4–25	L; C
TMP16 ^6^	BIM B-1571	ON248074	*Brachybacterium* *paraconglomeratum* LMG19861	99.93	5	10–37	Cas; A; U; BG; C
TMP21 ^7^	BIM B-1545	ON248079	*Micrococcus luteus* NCTC 2665	99.58	5	10–37	L; Cas; Gel; A; C
TMP23 ^9^	BIM B-1547	ON248081	*Agrococcus* *citreus* IAM 15145	99.50	18	18–25	C
TMP30 ^8^	BIM B-1567	ON248088	*Leifsonia* PHSC20C1	99.59	18	4–25	C
			*Leifsonia rubra* CMS 76R	99.45			

* MP—meltwater pond number, L—lipolytic activity (Twin 80), Cas—protease activity (Calcium–casein agar), Gel—protease activity (Gelatin), A—amylolytic activity, U—urease activity, D—DNase activity, BG—β-galactosidase (β-GAL) activity, C—catalase activity, ND—not determined.

## Data Availability

The data that support the findings of this study are available from the corresponding author upon reasonable request.

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
