# Peer review of "Isolation, Physiological Characterization, and Antibiotic Susceptibility Testing of Fast-Growing Bacteria from the Sea-Affected Temporary Meltwater Ponds in the Thala Hills Oasis (Enderby Land, East Antarctica)"

_biology, 2022, doi:10.3390/biology11081143_

Round 1

Reviewer 1 Report

The manuscript reports on the chatacterization of bacterial isolates from  sea-affected terrestrial temporary non-flowing meltwater Antarctic ponds. Overall, 29 isolates were obtained from 9 ponds, then screened for physiological and biochemical properties and identified by the 16rRNA sequencing. 

Despite the fact that it is the first time reporting the isolation of Antarctic bacteria from temporary ponds, that the manuscript is well written (even if it contains some typesetting errors), that methods are well described and chosen..., in my opinion 29 bacterial isolates are very few to justify the publication in Biology as they are not representative enough. Moreover, a unique culture medium was used for bacterial isolation, thus making a first selection of isolates. 

Author Response

Response to Reviewer 1 Comments

The manuscript reports on the characterization of bacterial isolates from sea-affected terrestrial temporary non-flowing meltwater Antarctic ponds. Overall, 29 isolates were obtained from 9 ponds, then screened for physiological and biochemical properties and identified by the 16rRNA sequencing. 

Point 1: Despite the fact that it is the first time reporting the isolation of Antarctic bacteria from temporary ponds, that the manuscript is well written (even if it contains some typesetting errors), that methods are well described and chosen..., in my opinion 29 bacterial isolates are very few to justify the publication in Biology as they are not representative enough. Moreover, a unique culture medium was used for bacterial isolation, thus making a first selection of isolates.

Dear Editor, thank you very much for your feedback. We would like to clarify that in a total of 92 bacterial isolates were isolated from the meltwater ponds, and out of these isolates for this study, we selected 29 with visually different phenotypic characteristics (colony shape and growth rate). Based on several similar studies published previously, we think that the number of isolates is sufficient for the manuscript to be published in Biology MDPI [1,2].

  1. Smirnova, M.; Tafintseva, V.; Kohler, A.; Miamin, U.; Shapaval, V. Temperature- and Nutrients-Induced Phenotypic Changes of Antarctic Green Snow Bacteria Probed by High-Throughput FTIR Spectroscopy. Biology 2022, 11, 890, doi:10.3390/biology11060890.
  2. Kanpiengjai, A.; Nuntikaew, P.; Wongsanittayarak, J.; Leangnim, N.; Khanongnuch, C. Isolation of Efficient Xylooligosaccharides-Fermenting Probiotic Lactic Acid Bacteria from Ethnic Pickled Bamboo Shoot Products. Biology 2022, 11, 638, doi:10.3390/biology11050638.

Reviewer 2 Report

Authors in the manuscript “Isolation, physiological characterization and antibiotic susceptibility testing of fast-growing bacteria from the sea-affected temporary meltwater ponds in the Thala Hills oasis (Enderby Land, East Antarctica)” identify and characterize culturable fast-growing bacteria isolated from the sea affected temporary meltwater ponds in the East Antarctica area of the Vecherny region of the Thala Hills Oasis, Enderby Land. The authors carry out a very complete, well-planned and justified study of the culturable fraction isolated from water samples.

These types of studies are of interest since they provide a battery of microorganisms with possible interest for the production of different valuable products..

The authors should define fast-growing bacteria.

The expression birds and animals is strange. Birds are animals: "animals including birds?"

Line 136. How do the authors prepare the samples for epifluorescence microscopy? Do they stain with DAPI or another fluorochrome? It must be explained.

Line 149, 285. What do authors refer to with growth rate? How do they determine it?

Line 211. At what temperature or temperatures are enzyme activity tests performed? It must be indicated.

Line 269. How do the authors calculate bacterial biomass? Should be explained in Material and Methods.

Figure 4. The range of colors should be explained.

Line 255. Change Daniela for Jara. Daniela is a first name.

Line 479. Change Carmen for Rizzo. Carmen is a first name.

Line 516. Change Tara for Connelly. Tara is a first name.

Author Response

Response to Reviewer 2 Comments

Point 1: The authors should define fast-growing bacteria.

Dear Reviewer, thank you for commenting on this. The definition of fast-growing bacteria was added to the manuscript. Due to the fact that the first isolation was performed for two weeks and all isolates were retrieved within this time and were able to grow within a range 2-6 days at optimal temperature we concluded that the isolated bacteria could be named as fast-growing. When defining the term ‘fast-growing bacteria’ we added several references where a similar definition was made. Please see lines 307-309.

Point 2: The expression birds and animals is strange. Birds are animals: "animals including birds?"

Thank you very much for noticing this mistake, we corrected this throughout the manuscript, please see lines 28-29,35,93,101,566-567,603.

Point 3: How do the authors prepare the samples for epifluorescence microscopy? Do they stain with DAPI or another fluorochrome? It must be explained.

Thank you very much for your very relevant comment. We added more clarifications on how the sample preparation was performed to the material and methods section 2.2. Bacterioplankton and physicochemical analysis of water samples. Please see lines 137-139.

Point 4. What do authors refer to with growth rate? How do they determine it?

The growth rate was determined visually through daily observation and selection of colonies appearing at various time points during the first 2-6 days of cultivation. Please see lines 167-169 for the clarifying sentence added to the materials and methods.

Point 5: At what temperature or temperatures are enzyme activity tests performed? It must be indicated.

Dear reviewer, this is specified in the material and methods section 2.6. Thermotolerance and enzymatic activity: “All plate-based assays were performed in duplicates at 18 °C for up to 10 days”(lines 240-241). In addition, we specify temperature in the main table with the results ( Table 2).

Point 6: How do the authors calculate bacterial biomass? Should be explained in Material and Methods.

We added more clarifications on how the sample preparation was performed to the material and methods  section 2.2. Bacterioplankton and physicochemical analysis of water samples. Please see lines 146-158.

Point 7: The range of colors should be explained.

Dear Reviewer, Thank you for your feedback; we have added more explanations to the figure for the color codes we used. We categorized colors into 3 groups: – phenotype (sensitive, intermediate, resistant, natural resistance), isolates that didn’t grow at a certain temperature and those for which data was not determined were also marked with the colors, temperature (18 and 25 °C) and groups of bacteria (Gram-negative and Gram-positive). We gave explanations for each color in each group.

Point 8: Line 255. Change Daniela for Jara. Daniela is a first name.

Please see line 708.

Point 9: Line 479. Change Carmen for Rizzo. Carmen is a first name.

Please see line 830.

Point 10: Line 516. Change Tara for Connelly. Tara is a first name.

Please see line 817.

Reviewer 3 Report

Akulava et al. presented the identification, enzymatic activity, and antibiotic sensitivity of bacterial isolates from the meltwater ponds in East Antarctica. This manuscript would gain interest to readers because it provides partial biochemical information of isolated psychrophilic microorganisms with potential biotechnological applications. In addition, their findings are also concerning particularly that many of these bacterial isolates are antibiotic-resistant, which may indicate that Antarctica is not that pristine as we previously thought. Although this manuscript contributes to environmental microbiology, there are some minor weaknesses that should be addressed prior to publication.

1. Line 98-100: The authors claimed that they are the first to isolate and characterize fast-growing bacteria from the meltwater ponds of East Antarctica. The authors should do extensive literature search and acknowledge/cite the contributions of other published studies on the microbial ecology of Antarctica. These previous reports can be incorporated in the introduction and discussion sections of the manuscript.

2. Table 2: The negative (-) sign under the enzymatic activity of C. funditum, C. iners, and F. tabacinasalis may connote that these isolates do not secrete hydrolytic enzymes. However, in line 351, the authors stated that these isolates “showed slow growth on minimal selective media,” which explains why no enzymatic activity was detected. Therefore, it would be more appropriate to indicate that the hydrolytic activities of these isolates were “not determined” instead of putting a negative (-) sign.

3. Line 358: “…proteolytic enzymes such as lactate dehydrogenase (LDHs)…” Lactate dehydrogenase is not a proteolytic enzyme. It converts lactate to pyruvate.

4. Figure 4: At the intersection of “TMP27” and “Total 18áµ’C,” the value is “0” but it appears that isolate TMP27 is resistant (or intermediately resistant) in some treatments at 18áµ’C.

5. Line 345: “…TMP23 did not grow at 10áµ’C but were able to grow at 37áµ’C…” However, in Table 2, it is indicated that TMP23 is thermotolerant at 18-25áµ’C.

Author Response

Response to Reviewer 3 Comments

Akulava et al. presented the identification, enzymatic activity, and antibiotic sensitivity of bacterial isolates from the meltwater ponds in East Antarctica. This manuscript would gain interest to readers because it provides partial biochemical information of isolated psychrophilic microorganisms with potential biotechnological applications. In addition, their findings are also concerning particularly that many of these bacterial isolates are antibiotic-resistant, which may indicate that Antarctica is not that pristine as we previously thought. Although this manuscript contributes to environmental microbiology, there are some minor weaknesses that should be addressed prior to publication.

Point 1: The authors claimed that they are the first to isolate and characterize fast-growing bacteria from the meltwater ponds of East Antarctica. The authors should do extensive literature search and acknowledge/cite the contributions of other published studies on the microbial ecology of Antarctica. These previous reports can be incorporated in the introduction and discussion sections of the manuscript.

Dear Reviewer, we totally agree with your comment, there are a few more studies done reporting on the microbial ecology of  East Antarctica. So in order to emphasize the uniqueness of our study, we specified the exact location of the meltwater ponds, since we are quite sure that no one reported the isolation of bacteria from the water bodies of this area. Please see ines 101-103.

Point 2: Table 2:The negative (-) sign under the enzymatic activity of C. funditum, C. iners, and F. tabacinasalis may connote that these isolates do not secrete hydrolytic enzymes. However, in line 351, the authors stated that these isolates “showed slow growth on minimal selective media,” which explains why no enzymatic activity was detected. Therefore, it would be more appropriate to indicate that the hydrolytic activities of these isolates were “not determined” instead of putting a negative (-) sign.

Thank you very much for finding this error. We made changes in Table 2 in accordance with your recommendation.

Point 3: “…proteolytic enzymes such as lactate dehydrogenase (LDHs)…” Lactate dehydrogenase is not a proteolytic enzyme. It converts lactate to pyruvate.

Thank you very much for noticing this mistake. The corrections are made, please see lines 385-389.

Point 4: Figure 4: At the intersection of “TMP27” and “Total 18áµ’C,” the value is “0” but it appears that isolate TMP27 is resistant (or intermediately resistant) in some treatments at 18áµ’C.

Thank you for noticing this mistake. We made changes in Figure 4 and added more clarification in material and methods section 2.7. Antibiotic Susceptibility Testing. For each strain the amount of antibiotics to which it is resistant (inhibition zones ≤ 15 mm) was counted. For each antibiotic the number of resistant strains was counted, separately for gram-positive and gram-negative bacteria and total amount. Please see lines 276-278.

Point 5: “…TMP23 did not grow at 10áµ’C but were able to grow at 37áµ’C…” However, in Table 2, it is indicated that TMP23 is thermotolerant at 18-25áµ’C.

Dear Reviewer, the corrections are made, please see lines 372-373.

Round 2

Reviewer 1 Report

I'm sorry but my scientific evaluation is the same as before.

I defer to the publisher's decision basing on the fact that papers with few strains have previously been published in Biology.